# Early-Life Colonization by Anelloviruses in Infants

**DOI:** 10.3390/v14050865

**Published:** 2022-04-22

**Authors:** Joanna Kaczorowska, Aurelija Cicilionytė, Anne L. Timmerman, Martin Deijs, Maarten F. Jebbink, Johannes B. van Goudoever, Britt J. van Keulen, Margreet Bakker, Lia van der Hoek

**Affiliations:** 1Amsterdam UMC, Laboratory of Experimental Virology, Department of Medical Microbiology and Infection Prevention, Location University of Amsterdam, Meibergdreef 9, 1105 AZ Amsterdam, The Netherlands; a.cicilionyte@amsterdamumc.nl (A.C.); a.l.timmerman@amsterdamumc.nl (A.L.T.); m.deijs@amsterdamumc.nl (M.D.); m.f.jebbink@amsterdamumc.nl (M.F.J.); m.e.bakker@amsterdamumc.nl (M.B.); 2Amsterdam Institute for Infection and Immunity, Postbus 22660, 1100 DD Amsterdam, The Netherlands; 3Amsterdam UMC, Department of Pediatrics, University of Amsterdam, Emma Children’s Hospital, Meibergdreef 9, 1105 AZ Amsterdam, The Netherlands; h.vangoudoever@amsterdamumc.nl (J.B.v.G.); b.j.vankeulen@amsterdamumc.nl (B.J.v.K.); 4Amsterdam Reproduction & Development Research Institute, Postbus 22660, 1100 DD Amsterdam, The Netherlands

**Keywords:** anelloviruses, *Anelloviridae*, breast milk, early-life infections, mother-to-child transmission, alphatorquevirus, betatorquevirus, gammatorquevirus

## Abstract

Anelloviruses (AVs) are found in the vast majority of the human population and are most probably part of a healthy virome. These viruses infect humans in the early stage of life, however, the characteristics of the first colonizing AVs are still unknown. We screened a collection of 107 blood samples from children between 0.4 and 64.8 months of age for the presence of three AV genera: the *Alpha*-, *Beta*- and *Gammatorquevirus*. The youngest child that was positive for AV was 1.2 months old, and a peak in prevalence (100% of samples positive) was reached between the twelfth and eighteenth months of life. Intriguingly, the beta- and gammatorqueviruses were detected most at the early stage of life (up to 12 months), whereas alphatorqueviruses, the most common AVs in adults, increased in prevalence in children older than 12 months. To determine whether that order of colonization may be related to oral transmission and unequal presence of AV genera in breast milk, we examined 63 breast milk samples. Thirty-two percent of the breast milk samples were positive in a qPCR detecting beta- and gammatorqueviruses, while alphatorqueviruses were detected in 10% of the samples, and this difference was significant (*p* = 0.00654). In conclusion, we show that beta- and gammatorqueviruses colonize humans in the first months of life and that breastfeeding could play a role in AV transmission.

## 1. Introduction

Anelloviruses (AVs) are small, circular single-stranded DNA viruses infecting vertebrates [1]. While found in basically all body compartments [2], AVs are considered one of the most common eukaryotic viruses in human blood [3]. AVs are widespread in the human population, however, the source and the moment of the first infection are still not clear [3,4]. Of note, higher levels of AV DNA are found in immunocompromised people, for instance, in HIV-1 positive subjects [5,6,7]. Since the HIV-1 infected individuals have significantly higher AV DNA prevalence and concentration compared to HIV-1 negative people, the transmission of AVs may be enhanced in these individuals [6].

Three AV genera are known to infect humans: *Alphatorquevirus* (often referred to as torque teno virus, TTV), *Betatorquevirus* (torque teno mini virus, TTMV), and *Gammatorquevirus* (torque teno midi virus, TTMDV). *Alphatorquevirus* is the most common in adults [8] and the most studied genus and contains 26 species, while *Beta*- and *Gammatorquevirus* contain 38 and 15 species that are officially approved as reference genomes by ICTV [9]. Even though they share similarities in genome organization, the three genera differ in genome size—3.8 to 3.9 kb for *Alphatorquevirus*, 3.2 kb for *Gammatorquevirus*, and 2.8 to 2.9 kb for *Betatorquevirus* [9]. Moreover, they are extremely divergent in terms of the genomic sequence [1,10].

AVs are nearly continuously shed by infants [11,12], indicating that first infections take place early in life. Umbilical cord blood is negative for alphatorquevirus DNA, thus transplacental transmission of AVs is considered unlikely [13]. Some studies have found alphatorquevirus DNA in blood as early as in the second month of life [14,15]. Of note, AVs are not only present in the blood, but also in stool samples collected in the first months of life, indicating replication in the gut at a very young age [12,16]. The peak of AV abundance in the gut was found between the sixth and twelfth months of life, after which the abundance decreases [12,16]. AVs may thus be transmitted by breastfeeding. Breast milk contains AV DNA [17,18], however, there was no association found between the breastfeeding status and AV richness in infants [19,20]. It has been shown that the human gut virome is highly dynamic in the first years of life [3,16]. We hypothesize that also the blood anellovirus virome (anellome [1]) may be unfixed in young infants. Early-life dynamics of anellome could be a part of the maturation of children’s immune systems [12,21]. To explore the hypothesis, we investigated the prevalence of three human-infecting AVs (alphatorquevirus, betatorquevirus, and gammatorquevirus) in the blood of 107 children between the age of 0.4 and 64.8 months using a set of three quantitative PCRs (qPCRs). Moreover, to investigate the significance of breastfeeding as an AV transmission route, we performed the same set of qPCRs on a collection of 64 breast milk samples. Additionally, we assessed the prevalence and viral load of AVs in 7 children’s serum and in 10 breast milk samples obtained from HIV-1 positive subjects.

## 2. Materials and Methods

### 2.1. Clinical Samples

Blood samples (serum or EDTA-plasma) were collected from 114 infants from 0.4 to 64.8 months of age (Appendix A). A total of 7 of the 114 blood samples were collected from HIV-1 infected children (CH108–CH114). Next to the blood samples, 73 breast milk samples were collected from other cohorts. Ten of these breast milk samples were collected from HIV-1-positive mothers (BM64–BM73) and 63 breast milk samples were the surplus of milk donated to the Dutch Human Milk Bank. The children and the women were not related to each other. The children’s serum samples and the HIV-1 positive breast milk samples had been anonymized. The Dutch Human Milk Bank samples were pseudo-anonymized and not traceable. Until nucleic acid isolations, the blood samples were stored at −80 °C, and the breast milk samples at −20 °C.

### 2.2. Nucleic Acid Isolations

One hundred µL of collected samples were lysed and the total nucleic acids were extracted using the Boom protocol [22]. This method is based on the lysing and nuclease-inactivating properties of the chaotropic agent guanidine thiocyanate (GuSCN) and the concurrent binding of all nucleic acid types by silica particles in the presence of this agent. Bound nucleic acids are subsequently washed and elution of nucleic acids is achieved by adding water (65 µL) to the silica particles. The nucleic acids were stored at −80 °C until further use.

### 2.3. Genus-Specific qPCRs

Three genus-specific qPCRs were performed to quantify the concentration of AV DNA in the samples. The first qPCR detects solely alphatorquevirus [23], the second detects only betatorquevirus, while the third qPCR detects both beta- and gammatorquevirus (primers are presented in Appendix A). Because of the high genetic divergence of AVs and the presence of only one sufficiently large conserved region in the genome, it was not feasible to design qPCR primers to detect solely the *Gammatorquevirus* genus. Dilutions of a positive control plasmid were used to construct the standard curve and calculate the sample concentrations of the viruses. The qPCR reaction mixture consisted of 2.5 µL of isolated nucleic acids, 6.25 µL of 2× Qiagen RotorGene Probe Master-mix (Qiagen, catalogue number 204574), 0.25 µL of probe, 0.5 µL of forward and 0.5 µL of reverse primer (all 10 µM) and 2.5 µL of H_2_O in each sample. The reaction was performed on a Rotor-Gene machine ( Qiagen GmbH, Hilden, Germany) as follows: 95 °C for 3 min, followed by 40 cycles of 95 °C for 3 s, 60 °C for 10 s, and a final elongation step at 72 °C for 3 min. The detection limit of the AV concentration was estimated experimentally and was 1.5 × 10^2^ copies per mL of the sample.

### 2.4. Statistical Analysis

All graphs and statistical analysis were performed in R version 4.1.1 equipped with the tidyverse, dplyr, and rstatix packages. The McNemar statistical test was used to compare the prevalence of the genera, the Chi-Square test to compare the AV prevalence in early and late breast milk, and the Wilcoxon sum rank test was used to compare DNA concentrations.

## 3. Results

### 3.1. Anellovirus Prevalence and DNA Concentration in Children’s Blood

Alphatorquevirus, betatorquevirus, and gammatorquevirus viral loads were quantified in 114 children’s serum (n = 112) or plasma (n = 2) samples. The children were between 0.4 to 64.8 months of age (Appendix A). Seven of the samples (CH108 to CH114) were collected from HIV-1 positive children and therefore were excluded from the overall AV prevalence analysis. Twenty-eight samples were negative in all three qPCRs. Forty-eight percent of the samples obtained from HIV-1-negative children were positive in the alphatorquevirus qPCR, 46% in the betatorquevirus qPCR, and 74% were positive in the beta + gammatorquevirus qPCR. The prevalence of Beta + Gammatorquevirus was significantly higher than that of alphatorquevirus (McNemar test, *p*-value = 4.89 × 10^−5^; Appendix A). A total of 30 out of 107 children’s blood samples (28%) were positive in all three qPCRs. All samples obtained from HIV-1 positive children (CH108 to CH114; Appendix A) were positive in all qPCRs and also showed the highest AV DNA concentrations (>10^7^ AV DNA copies per mL serum; Appendix A).

Next, we investigated the prevalence and concentration of AVs in relation to the age of the HIV-1 negative children. The prevalence of AVs increased with age, reaching a maximum of 100% at the age between 12 and 18 months (Appendix A). Moreover, the concentrations of AV DNA increased with the age of children and showed the highest values after 12 months of age (Appendix A), however, the only significant difference was observed between the subsets from 12 to 18 months and 30 to 65 months.

We observed a significantly higher (*p*-value = 0.00206) prevalence of beta- and gammatorquevirus than alphatorquevirus in the children of age between 0.4 and 12 months (Figure 1A and Appendix A). Between 0 and 12 months of age, less than half of the serum samples were positive for the alphatorquevirus qPCR (10 out of 50 samples, 20%). In contrast, the beta- and gammatorquevirus qPCRs were frequently positive in this age group (25 out of 50 samples, 50%). In children older than 12 months, the number of samples positive for alphatorquevirus DNA increases (41 out of 57 samples, 72%; Figure 1A,B).

Taken together, these results indicate that betatorqueviruses and/or gammatorqueviruses are the pioneer AVs infecting young children.

### 3.2. Prevalence of Anelloviruses in Breast Milk

Based on the hypothesis that breast milk could be the source of the first colonizing AVs in infants, we investigated a set of 73 breast milk samples. The samples were not related to the children’s samples. Ten of the breast milk samples were collected from HIV-1-positive women and thus were not included in the analysis of the overall prevalence of AVs in breast milk. A total of 24 out of 63 HIV-1 negative breast milk samples were positive for AV DNA (38% positive; Appendix A), with a substantial part of the samples qPCR positive in the Beta + Gammatorquevirus qPCR (32% positive samples), followed by Betatorquevirus qPCR (16% positive) and Alphatorquevirus qPCR (10% positive). The difference in the prevalence of Alphatorquevirus versus Beta + Gammatorquevirus was statistically significant (*p*-value = 0.00654; Appendix A). Six out of ten breast milk samples obtained from HIV-1 positive mothers were positive for AV DNA, five of the samples were positive in Alphatorquevirus, two in Betatorquevirus and six in Beta- and Gammatorquevirus qPCR (samples BM64–BM73; Appendix A).

We observed a similar AV DNA prevalence in early breast milk (collected between 0 and 2 months after an infant’s birth; 34% of the samples were positive for AV DNA) and late breast milk (after 2 months; 40% of samples were positive for AV DNA). Thus, the overall prevalence of AV in breast milk did not significantly change over time (Chi-square test, *p*-value = 0.8563; Figure 2, Appendix A).

In the breast milk samples, the mean DNA concentration measured in the Alphatorquevirus qPCR showed higher values than the other qPCRs, but this difference was not statistically significant (Wilcoxon test, *p* > 0.05). Of note are the high AV DNA concentrations measured in the breast milk samples obtained from three HIV-1 positive women: samples BM66, BM70, and BM71 contained >10^6^ AV DNA copies per mL of milk (Appendix A).

## 4. Discussion

In this study, we assessed the prevalence of AV in the blood of children of various ages and in breast milk. Most importantly, there was a higher prevalence of beta- and possibly also gammatorquevirus in the children between 0.4 and 12 months of age, compared to alphatorquevirus prevalence. We, therefore, propose that the alphatorquevirus is less likely to belong to the pioneer population of the children’s blood, but the *Beta-* and *Gammatorquevirus* do act more often as the first colonizers.

From 4 to 6 months after birth, most infants start with the initiation of solid food, next to breast milk (or formula). In this period, also the largest changes in bacteriome [24] and virome [16] are observed, which is connected with the cessation of the supply of breast milk-derived antibodies. In our dataset, the prevalence of alphatorquevirus indeed increased together with the age of children, and this genus was similarly common in children older than 12 months compared to the other two AV genera. *Alphatorquevirus* is the most common AV genus in adult blood [8,25]. We speculate that the blood anellome [1] may reach a compositional equilibrium around the age of 12 months.

There are a few hypothetical transmission routes of AVs: oral, airway, sexual transmission, or blood–blood contact [1,26]. Since the first infections happen early in life, it is most likely that the mother is the source of the pioneer viruses that populate the child. We detected AV DNA in nearly 40% of the tested breast milk samples. Thus, breastfeeding may play an important role in the transmission of AV from mother-to-child, which is in line with the findings reported by Ohto and colleagues [27]. More recent studies however showed that AVs are not particularly abundant in breast milk [17], and we also do not find high concentrations in the breast milk of healthy mothers; still, a low concentration may be enough to cause the colonization of AV in infants. In our dataset, the beta- and/or gammatorquevirus were the most prevalent in children’s blood samples. We observed the same unequal presence in breast milk, since 32% of breast milk samples were positive for beta- and/or gammatorquevirus, and this prevalence was significantly higher than the one of alphatorquevirus (10%). However, it is important to mention that the breastfeeding status of the tested children was unknown. Therefore, we propose that transmission via breast milk might be an important but not sole source of pioneer AVs in children; however, more research is needed to further support this claim.

It is of note, however, that we observed higher AV DNA concentrations in the blood of HIV-1-infected children and the breast milk of HIV-1-positive women compared to samples derived from HIV-1 negative subjects. It has been reported that individuals infected by HIV-1 tend to have higher levels of AV [5,7] and that there is a positive correlation between the levels of immunosuppression and AV abundance [28]. The HIV-1 positive children from our dataset were not breastfed, but still contained high levels of AVs in their blood already at 2 months of age. Due to the lack of breastfeeding, the AV transmission in these children must have happened via a different route. A metagenomic study involving serum, stool, nasopharyngeal swabs, and breast milk samples from mothers together with an analysis of longitudinally collected serum samples of their infants may not only provide information on AV infection compartmentalization but also reveal the routes that transmit AVs.

## Figures and Tables

**Figure 1 viruses-14-00865-f001:**
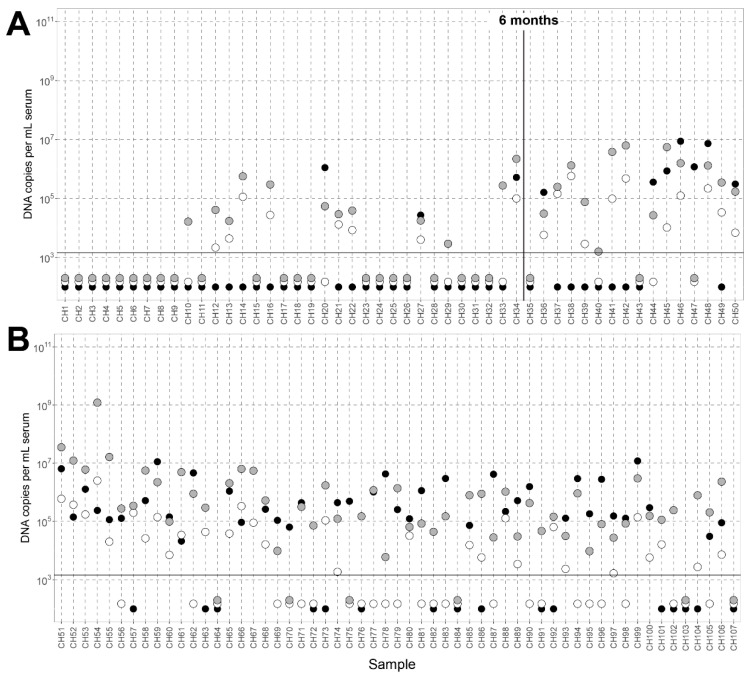
Anellovirus concentration in the children’s blood over months. (**A**) Anellovirus concentration in the first 12 months of life and (**B**) anellovirus concentration in children aged 12 months until 65 months. All samples are arranged in the order of children’s age and the age of 6 months is depicted with a vertical line. The samples under the horizontal line were either negative for the viruses, or samples contained concentrations below the cutoff. Filled black dots represent *alphatorquevirus* white dots betatorqueviruses, and gray dots gamma- or betatorqueviruses.

**Figure 2 viruses-14-00865-f002:**
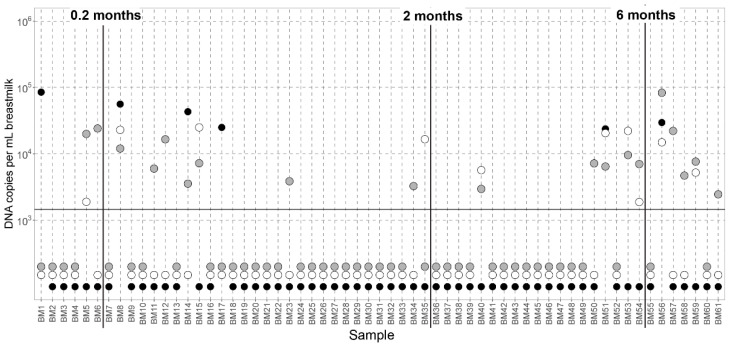
Anellovirus concentration in breast milk. All samples are arranged in the order of children’s age at the moment of the breast milk collection. The samples under the horizontal line were either negative for the viruses, or samples contained concentrations below the cutoff. Filled black dots represent alphatorqueviruses, white dots betatorqueviruses, and gray dots gamma- or betatorqueviruses.

## Data Availability

All generated data (qPCR results) are included in the article and in the Appendix A. The R code that was used to generate figures and perform the statistical analysis is available at a GitHub repository under the following link: https://github.com/joannakaczorowska/firstcolonizer (accessed on 20 April 2022).

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
