# Peer review of "Early-Life Colonization by Anelloviruses in Infants"

_viruses, 2022, doi:10.3390/v14050865_

Round 1

Reviewer 1 Report

The authors show the dynamics of anelovirus colonisation in the first months/years of life of children. The paper is well written, the analyses are well designed and the results are convincing. My main concern is the inclusion of some HIV-1 positive individuals, as it is well known that immune status influences viral load. Therefore, I believe that excluding these data from the overall analyses will increase the consistency/homogeneity of the study population and will not affect the overall results. My suggestion is not to completely eliminate these data, but to use them for comparison, despite their small sample size.

Some additional minor comments are:

  • According to the International Committee on Taxonomy of Viruses (ICTV), TTV, TTMV, and TTMDV have been recently subdivided into 26, 38, and 15 species. These figures are different from those stated by the authors in the introduction.
  • I asume that it not feasible to design qPCR primers for exclusively detecting TTMDVs, but this should be acknowledged by the authors
  • In figure 2, BM61 sample is missing. I don’t understand why

Author Response

The authors show the dynamics of anelovirus colonisation in the first months/years of life of children. The paper is well written, the analyses are well designed and the results are convincing. My main concern is the inclusion of some HIV-1 positive individuals, as it is well known that immune status influences viral load. Therefore, I believe that excluding these data from the overall analyses will increase the consistency/homogeneity of the study population and will not affect the overall results. My suggestion is not to completely eliminate these data, but to use them for comparison, despite their small sample size.

Response: We would like to thank the reviewer for this suggestion. We removed the data obtained from HIV-1-positive subjects from the overall results, and now present them just to compare the anellovirus prevalence and viral loads in the HIV-1 positive versus HIV-1 negative subjects (the comparison is shown in Supplementary Figure S2). We now mention the exclusion of the HIV-1 positive samples from the overall data in the results section, and Figure 1, Supplementary Figure S1 and Supplementary Table S3 were updated accordingly.

Some additional minor comments are:

  • According to the International Committee on Taxonomy of Viruses (ICTV), TTV, TTMV, and TTMDV have been recently subdivided into 26, 38, and 15 species. These figures are different from those stated by the authors in the introduction.

Response: We would like to thank the reviewer for the remark. The figures are now corrected in the manuscript and the most recent reference describing the updated taxonomy of anelloviruses is now cited.

  • I asume that it not feasible to design qPCR primers for exclusively detecting TTMDVs, but this should be acknowledged by the authors

Response: Yes – it was indeed not feasible to design TTMDV primers that detected solely TTMDV genus. A short paragraph is now added in the materials and methods section (lines 94 – 97 in the manuscript file with track changes).

  • In figure 2, BM61 sample is missing. I don’t understand why

Response: Apologies, the BM61 sample is now included in the graph.

Reviewer 2 Report

Early-life colonization by anelloviruses in infants by Kaczorowska et al. is a concise manuscript on a topic of some interest but with limited findings. I have only very minor comments.

I would suggest introducing the significance of HIV status of infant and mother within the introduction as this is discussed within the methods, results and discussion and does have an impact on your findings.

Line 75: Is it possible to give a short description of this protocol?

Author Response

Early-life colonization by anelloviruses in infants by Kaczorowska et al. is a concise manuscript on a topic of some interest but with limited findings. I have only very minor comments.

I would suggest introducing the significance of HIV status of infant and mother within the introduction as this is discussed within the methods, results and discussion and does have an impact on your findings.

Response: We thank the reviewer for the suggestion. A short mention on HIV-1 infection status influence on AV prevalence and load is now added to the introduction.

Line 75: Is it possible to give a short description of this protocol?

Response: A short description of the Boom extraction protocol was added to the materials and methods section.